# The Gene Structure and Expression Level Changes of the *GH3* Gene Family in *Brassica napus* Relative to Its Diploid Ancestors

**DOI:** 10.3390/genes10010058

**Published:** 2019-01-17

**Authors:** Ruihua Wang, Mengdi Li, Xiaoming Wu, Jianbo Wang

**Affiliations:** 1State Key Laboratory of Hybrid Rice, College of Life Sciences, Wuhan University, Wuhan 430072, China; wangruihua@whu.edu.cn (R.W.); mengdili@whu.edu.cn (M.L.); 2Oil Crops Research Institute of the Chinese Academy of Agricultural Sciences, Key Laboratory of Biology and Genetic Improvement of Oil Crops, Ministry of Agriculture, Wuhan 430072, China; wuxm@oilcrops.cn

**Keywords:** *GH3* gene, *Brassica napus*, orthologous gene, cis-element, expression pattern, allopolyploidization

## Abstract

The *GH3* gene family plays a vital role in the phytohormone-related growth and developmental processes. The effects of allopolyploidization on *GH3* gene structures and expression levels have not been reported. In this study, a total of 38, 25, and 66 *GH3* genes were identified in *Brassica rapa* (A_r_A_r_), *Brassica oleracea* (C_o_C_o_), and *Brassica napus* (A_n_AC_n_C_n_), respectively. *BnaGH3* genes were unevenly distributed on chromosomes with 39 on A_n_ and 27 on C_n_, in which six *BnaGH3* genes may appear as new genes. The whole genome triplication allowed the *GH3* gene family to expand in diploid ancestors, and allopolyploidization made the *GH3* gene family re-expand in *B. napus*. For most *BnaGH3* genes, the exon-intron compositions were similar to diploid ancestors, while the cis-element distributions were obviously different from its ancestors. After allopolyploidization, the expression patterns of *GH3* genes from ancestor species changed greatly in *B. napus*, and the orthologous gene pairs between A_n_/A_r_ and C_n_/C_o_ had diverged expression patterns across four tissues. Our study provides a comprehensive analysis of the *GH3* gene family in *B. napus*, and these results could contribute to identifying genes with vital roles in phytohormone-related growth and developmental processes.

## 1. Introduction

Polyploidization, as a prevalent phenomenon, is an important method of speciation in the plant kingdom. In angiosperms, approximately 70 percent of species have undergone one or several rounds of polyploidization in the evolutionary process [1]. Allopolyploids are derived from the hybridization between different species, followed with chromosome doubling or the fusion of unreduced gametes between different species. Allopolyploids usually have better adaptability and superior traits than their parents, which may contribute to natural selection and crop domestication [2]. Extensive changes on genetic and gene expression levels have been reported in allopolyploids. Genetic changes contain chromosomal rearrangements, DNA sequence elimination, and amplification. Chromosomal rearrangements could lead to the variation on seed yields and flowering time in *Brassica* allotetraploids [3,4]. Comparative analysis of gene structure among polyploid wheat and model grass genomes reveals that many genes have exonic sequence acquirement and loss, and 35 percent of these gene structure rearrangements give rise to premature termination codons and frame-shift mutations in polyploid wheat [5]. In addition, allopolyploidization has a profound impact on gene expression levels, including gene silencing, gene activation, gene non-additive expression, and gene bias expression. The gene expression levels of hexaploid wheat are compared with its parents, which reveals that the gene silencing is caused by gene regulation rather than gene loss, and one gene was specifically activated, suggesting that gene expression changes are correlative with polyploidization [6]. Non-additive gene expression has been found in diverse polyploids and is associated with allopolyploid growth vigor [7]. Genome-biased alterations in gene expression may be a way to overcome incompatibilities after different genome fusion, thus might facilitate the rapid genetic diploidization of allopolyploids [8]. In conclusion, changes on genetic and gene expression levels could contribute to adaptive enhancement during the evolution of allopolyploids.

Genes rapidly and transiently induced by auxin are regarded as early/primary auxin-responsive genes, which mainly belong to three gene families, including auxin/indole-3-acetic acid (*Aux/IAA*), Gretchen Hagen3 (*GH3*), and small auxin-up RNA (*SAUR*) gene family [9]. The GH3 proteins catalyze conjugation of amino acids with free indole-3-aceticacid (IAA), jasmonic acid (JA), and salicylic acid (SA) to modulate the concentrations of hormone bioactive forms during plant development and stress adaptation processes [10]. The GH3 proteins are classified into three groups according to protein sequence similarities and substrate specificities in *Arabidopsis thaliana* [11,12]. Group I is made up of two members: AtGH3.11, which adenylates jasmonic acid (JA) and acts as JA-amido synthetase, and AtGH3.10, which does not own these features [13]. Group II, containing eight members, adenylates indoleacetic acid (IAA) and conjugates IAA to amino acids. In addition, AtGH3.5 in group II also adenylates salicylic acid (SA) and displays SA-amino synthetase activity. Group III consists of nine members. AtGH3.12/PBS3 in group III conjugates glutamic acid (Glu) to 4-substituted benzoates, which is involved in signal transduction of SA [14], and the function of the other eight members is still unknown. The auxin-responsive elements (AuxREs) that possess core motifs (TGTCTC) exist in the *GH3* gene promoters, and the AuxRE motifs are various either in sequences or number [10]. Besides, the other hormone-related elements and stress-related elements are also found in the promoter regions of the *GH3* genes, which ensures their fast and transient responses to the other hormones and stress stimulus [15,16,17]. The GH3 proteins play important roles in plant development and light signal pathways [18,19,20,21]. So far, the identification and analysis of the *GH3* gene family have been performed in different plant species, such as *Arabidopsis thaliana*, rice, maize, *Medicago truncatula*, *Carica papaya*, and so on [12,17,22,23,24].

The divergence between *Brassica* and Arabidopsis is estimated to be 14.5 to 20.4 million years ago (MYA) [25]. The *Brassica* genome went through the whole genome triplication (WGT) about 15.9 MYA, followed by species divergence between *B. rapa* (A_r_A_r_, 2n = 20) and *B. oleracea* (C_o_C_o_, 2n = 18) about 4.6 MYA [26]. *B. napus* (A_n_A_n_C_n_C_n_, 2n = 38) is a natural allotetraploid, formed by the hybridization between *B. rapa* and *B. oleracea* followed with chromosome doubling [27]. Some GH3 proteins are found to have crucial roles in *Brassica* plant organ development and stress responses. A few GH3 proteins in *B. napus* are involved in seedling development, controlling leaf morphology and branch angle regulation [28,29,30]. The *BrGH3.12* that is highly expressed in the leaf apical region regulates heading type in *B. rapa* [31]. One *GH3* gene in *Brassica juncea* is strongly induced by Cd in roots and leaves [32]. In addition, a GH3 protein in *B. rapa* takes part in resistance to drought stress [33]. To date, very little information about the *GH3* gene family in *B. napus* is known. It is unknown how the gene structures and expression levels of the *GH3* gene family change in *B. napus* compared with diploid ancestors. The availability of *Brassica* genome annotations provides an opportunity for genome-wide analysis of the *GH3* gene family, which contributes to a better understanding of the function and evolution of the *GH3* gene family in *B. napus*.

In this study, the genome-wide analysis of the *GH3* gene family in *B. napus* were carried out. This study aimed to address: (1) the identification and characteristics of *GH3* genes in the three species of *Brassica*; (2) the source analysis of the *GH3* gene family members in *B. napus*; (3) the comparative analysis of the *GH3* gene exon-intron organization, cis-element distribution, and expression patterns between *B. napus* and ancestor species. The comprehensive analysis of the *GH3* gene family would provide a better understanding of the *GH3* gene function in polyploid growth and development, and a reference for exploring the adaption and evolution mechanisms of polyploid.

## 2. Materials and Methods

### 2.1. Plant Materials

Plant materials contained *B. rapa* cv. Chiifu (A_r_A_r_, 2n = 20), *B. oleracea* cv. Jinzaosheng (C_o_C_o_, 2n = 18), and *B. napus* cv. Darmor (A_n_A_n_C_n_C_n_, 2n = 38). The seeds of the three species of *Brassica* were acquired from the Oil Crops Research Institute of the Chinese Academy of Agricultural Sciences, China. Plant materials were planted in large flowerpots outside the greenhouse of Wuhan University, Wuhan (30.52 degrees north latitude and 114.31 degrees east longitude), China, under natural conditions. Plant materials were planted in September (autumn), and plant tissues were taken in March of the following year (spring).

### 2.2. *GH3* Gene Retrieval in the Three Species of *Brassica*

*Brassica rapa* (v1.5), *B. oleracea* (v1.1), and *B. napus* (v5.0) genome annotation files and protein sequences were downloaded from BRAD database (http://brassicadb.org/brad/). A total of 19 GH3 protein sequences in *A. thaliana* were obtained from TAIR database (https://www.arabidopsis.org/). All protein sequences of each *Brassica* species were searched via BLASTP algorithms with E values lower than 1e-10, using 19 GH3 protein sequences of *A. thaliana* as queries. Further, the Hidden Markov Model (HMM) profile of GH3 domain (Pfam: 03321) was used to search all protein sequences of each *Brassica* species using HMMER3 software. For each *Brassica* species, all GH3 protein candidates gained from these two searches were combined to make a non-redundant protein list. Finally, all non-redundant protein sequences were analyzed to confirm the presence of the entire GH3 domain by PHMMER (http://plants.ensembl.org/hmmer/index.html), InterProScan (http://www.ebi.ac.uk/interpro/sequence-search), and NCBI Conserved Domain Search (https://www.ncbi.nlm.nih.gov/Structure/cdd/wrpsb.cgi). The *GH3* genes in the three species of *Brassica* were renamed on the basis of the standard gene nomenclature for the *Brassica* genus [34].

### 2.3. Chromosome Location, Gene Structure, and Protein Property Analysis

The information about the chromosomal locations of the *GH3* genes were obtained from GFF files in BRAD database. The diagrams of the gene chromosome locations were drawn using the MapInspect software. The structures of the *GH3* genes were displayed to illustrate the exon-intron composition by TBtools according to the gene annotation from GFF files. The GH3 protein properties, including molecular weight (MW), isoelectric points (pI), and grand average of hydropathy (GRAVY) were calculated by ProtParam tool (https://web.expasy.org/protparam/). The sequences of all GH3 proteins were carried on multiple amino acid alignment by MAFFT to find conserved regions. The protein structures of AtGH3-5 (5KOD), AtGH3-11 (4EPL), AtGH3-12 (4EWV), and AtGH3-15 (6AVH) were obtained from Protein Data Bank (https://www.rcsb.org/). The online tool Phyre2 (http://www.sbg.bio.ic.ac.uk/phyre2/html/page.cgi?id=index) was used to predict the GH3 protein dimensional structure via homology modeling under “intensive” modelling mode [35]. The software Swiss-PdbViewer_4.10 was used to examine the locations of α-helices/β-strands and display the protein dimensional structure.

### 2.4. Identification of Orthologous Genes and Syntenic Genes Among the Three Species of *Brassica* and *Arabidopsis thaliana*

The most orthologous sets of *GH3* genes among *B. rapa*, *B. oleracea*, *B. napus*, and *A. thaliana* were acquired from the previous study [27], and others were identified by reciprocal BLASTP analysis. Syntenic genes were searched using online tool SynOrths (http://brassicadb.org/brad/searchSyntenytPCK.php) [36]. The Syntenic relationships were shown by Circos [37].

### 2.5. Analysis of Duplication Pattern and Phylogenetic Relationship of the *GH3* Gene Family

Five types of gene duplications were examined using MCScanX, namely singleton, dispersed, proximal, tandem, and segmental duplication/WGD [38]. Phylogenetic tree was constructed by MEGA7 with the neighbor-joining (NJ) method and bootstrap replication of 1000.

### 2.6. Calculation of d_N_ and d_S_ Values

The protein sequences of each GH3 orthologous pair between *B. napus* and two diploid ancestors were aligned using MAFFT. Then, a multiple sequence alignment of proteins and the corresponding DNA sequences were inputted into PAL2NAL to convert into a codon alignment. Finally, the codon alignment was subjected to the calculation of synonymous (d_S_) and non-synonymous (d_N_) substitution rates by the codeml program in PAML [39].

### 2.7. Search of cis-Elements in the *GH3* Gene Promoter Regions

For all *GH3* genes, the 2000 bp of upstream genomic sequences relative to the translation start codon were extracted from the three species of *Brassica* genomes. PLACE was used to scan cis-elements on both strands of the promoter sequences [40].

### 2.8. *GH3* Gene Expression Analysis

Stems, young leaves, blooming flowers, and siliques (ten days after pollination) were collected from nine independent plants (three *B. rapa*, three *B. oleracea*, and three *B. napus*) that grew for six months. There were three biological replicates for stems, leaves, flowers, and siliques in each *Brassica* species. All plant tissues were frozen in liquid nitrogen for later use. Total RNA was extracted from these plant tissues using Trizol reagent (Invitrogen) with the manufacturer’s procedure. The residual DNA was digested using RNase-free DNase I (Fermentas, Canada). The thirty-six cDNA libraries were constructed, and then checked for quality using ABI StepOnePlus Real-Time PCR System Qualification and Agilent 2100 Bioanalyzer. Finally, 150 bp paired-end sequencing was performed using an Illumina HiSeq XTen. All reads were qualified: the adapter reads, the reads in which the proportion of unidentified base was greater than five percent, and the low-quality reads, in which proportion of base (Q ≤ 15) was greater than twenty percent were removed. The clean reads of the three species of *Brassica* were mapped to the *B. rapa*, *B. oleracea*, and *B. napus* reference genome sequences using Bowtie2, respectively [41]. The reads that mapped to reference genome were assembled into the transcripts using Cufflinks software. The Cuffcompare program was used to annotate the assembled transcripts according to the annotation files of the three species genome sequence. Next, RSEM was used to calculate the expression levels of *GH3* genes [42]. The measurement unit was fragments per kilobase of exon per million mapped reads (FPKM). *Z*-value was used to normalize expression levels, and the FPKM values of all *GH3* genes were converted to *Z*-values, then *Z*-values were used to construct the heatmaps by TBtools. $Z-\mathrm{value}\;=\frac{\log2(\mathrm{FPKM})-\mathrm{Mean}(\log2(\mathrm{FPKM})\;\mathrm{of}\;\mathrm{all}\;\mathrm{samples})}{\mathrm{standard}\;\mathrm{deviation}\;\left(\log2\left(\mathrm{FPKM}\right)\;\mathrm{of}\;\mathrm{all}\;\mathrm{samples}\;\right)}$. The raw data of RNA-seq reads were deposited in the NCBI database (accession number SRR7816633-SRR7816668, unreleased).

## 3. Results

### 3.1. Identification and Chromosomal Distribution of *GH3* Genes

A total of 38, 25, and 66 *GH3* genes were confirmed in *B. rapa*, *B. oleracea*, and *B. napus*, respectively (Table 1). Each *GH3* gene was allocated a unique name in accordance with the standardized gene nomenclature (Appendix A). Compared with *A. thaliana*, *Brassica* genome experienced a unique triplication, so each *AtGH3* gene theoretically had three copies in *B. rapa* and *B. oleracea*. However, that is not the case. The *AtGH3-4* and *AtGH3-13* genes had no copy in *B. rapa* and *B. oleracea*. Most *AtGH3* genes had one or two copies in *B. rapa* and *B. oleracea*. This indicated that *B. rapa* and *B. oleracea* underwent gene loss in the process of *Brassica* WGT (whole genome triplication) after the divergence with *A. thaliana*. In addition, *AtGH3-8*, *AtGH3-12*, and *AtGH3-19* had more than three copies in *B. rapa*, and *AtGH3-8* also had more than three copies in *B. oleracea*, suggesting that a few *GH3* genes underwent gene expansion in *B. rapa* and in *B. oleracea*. *B. napus* is derived from the hybridization between *B. rapa* and *B. oleracea* followed with chromosome doubling, and the number of *GH3* genes in *B. napus* should be the sum of two diploids in theory. However, *BnaGH3* also had expansion or loss after allopolyploidization. For example, the number of *BnaGH3-8* genes exceeded the sum of *BraGH3-8* and *BolGH3-8* genes, while the number of *BnaGH3-19* genes was one third of the total number of *BraGH3-19* and *BolGH3-19*. Although *BnaGH3* genes went through expansion or loss during the evolution process, the number of *BnaGH3* genes was close to the sum of *BraGH3* and *BolGH3* genes.

The *GH3* genes were unevenly distributed across the chromosomes in *B. napus*. The number of *BnaGH3* genes in the A_n_ and C_n_ subgenomes was 37 and 29, respectively, which was similar to that in *B. rapa* (A_r_, 38) and *B. oleracea* (C_o_, 25). The number distribution comparison of *GH3* genes between A_n_ and A_r_, C_n_, and C_o_ was carried out. The number of *GH3* genes on each chromosome was counted, and the results were shown in Table 2. For the A_n_ subgenome, the top three chromosomes were A_n_03, A_n_09, and A_n_06 in *B. napus*, which was the same as ancestor *B. rapa*, in which the top three chromosomes were A_r_03, A_r_09, and A_r_06. The number distribution pattern of the remaining *BnaGH3* genes in the A_n_ subgenome was 4 (A_nn_-random), 3 (A_n_05), 2 (A_n_01, A_n_08 and A_n_10), 1 (A_n_02, A_n_04-random, A_n_07, and A_n_09-random). The number distribution pattern of the remaining *BraGH3* genes in the A_r_ genome also was 4 (A_r_02 and scaffords), 3 (A_r_10), 2 (A_r_04 and A_r_07), 1 (A_r_01, A_r_05, A_r_08). For the C_n_ subgenome, C_nn_-random contained the largest number. However, for the C_o_ genome, C_o_03 and C_o_09 contained the largest numbers. There was no *BnaGH3* gene on C_n_05, which was the same as C_o_05. Excluding C_n_02, the number distribution pattern of the remaining *BnaGH3* genes in the C_n_ subgenome was still 4 (C_n_09), 3 (C_n_01, C_n_03), 2 (C_n_04, C_n_07 and C_n_08), 1(C_n_06), which was in line with the C_o_ genome. Chromosomal location distribution of *GH3* genes in the three species of *Brassica* was shown in Figure 1.

### 3.2. Orthologous Relationship and Synteny Analysis of *GH3* Genes

Thirty-three orthologous *GH3* gene pairs were identified between A_n_ and A_r_, and twenty orthologous *GH3* gene pairs were found between C_n_ and C_o_ (Appendix A). Chalhoub et al. reported that 27,360 of 32,699 orthologous gene pairs (83.7%) between ancestor *B. rapa* and *B. oleracea* were retained as homologous gene pairs in *B. napus* [27]. Our results revealed that most of the orthologous *GH3* gene pairs (82.6%) between *B. rapa* and *B. oleracea* were retained as homologous gene pairs in *B. napus* (Appendix A). These results suggested that most of the *GH3* genes remained intact during the formation and evolution of allotetraploid *B. napus*.

Thirteen *BnaGH3* genes had no corresponding *BraGH3* or *BolGH3* orthologous genes (four *BnaGH3* genes on the A subgenome had no corresponding *BraGH3* orthologous gene, and nine *BnaGH3* genes on the C subgenome had no corresponding *BolGH3* orthologous gene in Appendix A), and we analyzed the origins of these thirteen genes. Seven *BnaGH3* genes (Appendix A) had corresponding *B. rapa* or *B. oleracea* orthologous genes, but the proteins that these ancestral orthologous genes corresponded to did not have a complete GH3 domain. There may have been two possibilities. In the first possibility, the sequences of the *GH3* genes from the ancestors changed after allopolyploidization, which resulted in the formation of a complete GH3 domain. In the second possibility, before the occurrence of allopolyploidization, the proteins that the seven ancestral genes corresponded to had complete GH3 domains. After allopolyploidization, the proteins that the seven genes from the ancestors corresponded to still had complete GH3 domains in *B. n**apus*, and the GH3 domains had been kept complete. However, for the ancestors, the proteins that the seven genes corresponded to lost the complete GH3 domains in the recent evolutionary process. The remaining six *BnaGH3* genes (BnaC02.GH3-8.b, BnaC02.GH3-8.c, BnaA09.GH3-8.g, BnaCX.GH3-8.g, BnaAX.GH3-8.i, and BnaCX.GH3-17.c) had no corresponding *B. rapa* or *B. oleracea* orthologous genes, indicating that these six genes may appear as new genes during the formation of *B. napus*. In the next section, we carried out the analysis of the gene duplication, and BnaA09.GH3-8.g, BnaC02.GH3-8.b, and BnaC02.GH3-8.c were newly generated tandem repeat genes after allopolyploidization.

Syntenic genes are orthologous genes that locate in syntenic fragments between different species that derive from a shared ancestor. They usually have similar functions, so syntenic gene analysis is used to understand gene function of newly sequenced genomes. Besides, syntenic gene analysis can also reveal species genomic evolution. Three subgenomes are produced because of genome triplication in the diploid *B. rapa* and *B. oleracea*, and then a lot of genes are lost. According to the extent of gene retention, three subgenomes are specified as LF (maximum gene retention), MF1, and MF2 (least gene retention). The results of syntenic gene analysis showed that *AtGH3-11* had three syntenic orthologs in both *B. rapa* and *B. oleracea*, which confirmed the occurrence of *Brassica* genome triplication. *AtGH3-12* also had three syntenic orthologs in *B. rapa*. The other *AtGH3* genes had 0, 1, or 2 syntenic orthologs in *B. rapa* or *B. oleracea*, which suggested that *GH3* genes underwent loss after *Brassica* genome triplication. After allopolyploidization, each *AtGH3* gene theoretically had six syntenic orthologs in *B. napus*, but only *AtGH3-12* had six syntenic orthologs in *B. napus*. The other *AtGH3* genes had zero to four syntenic orthologs in *B. napus*. There were three ways to lose syntenic *BnaGH3* orthologs. First, after the *Brassica* genome triplication, the loss of the *GH3* genes in the diploid *B. rapa* and *B. oleracea* resulted in the loss of syntenic *BnaGH3* orthologs. For example, *AtGH3-3* had one syntenic ortholog in both *B. rapa* and *B. oleracea*, so it had two syntenic *BnaGH3* orthologs. Second, the loss of the *GH3* genes after the allopolyploidization. For example, *AtGH3-11* had three syntenic orthologs in both *B. rapa* and *B. oleracea*, while it had three syntenic orthologs in *B. napus*. Finally, the *Brassica* genome triplication process and the allopolyploidization process together caused the loss of syntenic *BnaGH3* orthologs. For example, *AtGH3-1* had one syntenic ortholog in both *B. rapa* and *B. oleracea*, while it had no syntenic ortholog in *B. napus*. In addition, allopolyploidization could make *AtGH3* genes gain syntenic *BnaGH3* orthologs. *AtGH3-9* had no syntenic ortholog in both *B. rapa* and *B. oleracea*, while it had two syntenic orthologs in *B. napus*. *AtGH3-12* had three and two syntenic orthologs in *B. rapa* and *B. oleracea*, while it had six syntenic orthologs in *B. napus*. The syntenic relationship pairs between *GH3* genes from the three species of *Brassica* and *A. thaliana* were shown in Figure 2.

To comprehend whether natural selection acted on the evolution of the *GH3* gene family in *B. napus*, selection pressure analysis was performed on the orthologous *GH3* gene pairs between A_n_ and A_r_, C_n_, and C_o_. The non-synonymous rate (d_N_) and synonymous rate (d_S_) value were calculated. The d_N_/d_S_ ratio > 1 represents positive selection, the d_N_/d_S_ ratio = 1 represents neutral selection, and the d_N_/d_S_ ratio < 1 represents purifying selection [43]. The d_N_/d_S_ ratios of the orthologous gene pairs were shown in Appendix A. Seven *BnaGH3* genes had d_N_/d_S_ ratios that were >>1, suggesting that these genes experienced stronger positive selection pressure during the evolution process. In addition, the d_N_/d_S_ ratios of BnaC09.GH3-10.a and BnaC06.GH3-7.a were 1.3000 and 1.6412, respectively, which indicated that these two genes were subject to positive selection pressure. The positive selection could promote the function changes of these nine *BnaGH3* genes to survive, and mutations were advantageous. The rest of *BnaGH3* genes had d_N_/d_S_ ratios that <1, which indicated that they underwent purifying selection during the evolution process and may preferentially conserve function, with mutations being disadvantageous. The average d_N_/d_S_ ratio of *BnaGH3* genes on the A_n_ subgenome was higher than that on the C_n_ subgenome, showing that *BnaGH3* genes on the A_n_ subgenome underwent higher selection pressure during the evolution process of *B. napus*.

### 3.3. The Duplication Pattern Analysis of *GH3* Genes

The duplication patterns of *GH3* genes were analyzed in *B. napus* and two ancestor *Brassica* species by MCScanX. The four duplicated types were found in each *Brassica* species, namely dispersed, proximal, tandem, and segmental duplication/WGD (Appendix A). We found that 55% of *BraGH3* genes, 72% of *BolGH3* genes, and 62% of *BnaGH3* genes were generated by WGD, while 26% of *BraGH3* genes, 12% of *BolGH3* genes, and 24% of *BnaGH3* genes were generated by tandem duplication, suggesting that the duplication pattern of *GH3* genes was high WGD and low tandem duplication. The previous studies showed that WGD contributed the most to the expansion of the gene families in *B. napus* [44,45]. Our findings were consistent with the previous studies, and the WGD type had the largest number of *GH3* genes, which indicated that the main cause of the expansion of *GH3* genes was WGD. The ancestral *Brassica* genome underwent a triplication event, which could greatly promote the expansion of the *GH3* gene family in *B. rapa* and *B. oleracea*. Next, *B. napus* was formed by the hybridization between *B. rapa* and *B. oleracea* followed with chromosome doubling. Therefore, the expansion of the *GH3* gene family in *B. napus* was affected by *Brassica* WGT and allopolyploidization. Four, one, and five clusters of *GH3* tandem repeat genes were found in *B. rapa*, *B. oleracea*, and *B. napus*, respectively (Table 3). Three clusters of tandem repeat genes from *B. rapa* were completely retained in *B. napus*. Although BnaA09g48090D was the orthologous gene of BraA09.GH3-19.d, it was not a member of the *GH3* gene family, because BnaA09g48090D protein had a GH3 domain with the missing N-terminal, so it was not in Table 3. The orthologous gene of BraA06.GH3-8.f was BnaA06.GH3-8.e. Due to the change in the position of BnaA06.GH3-8.e after allopolyploidization, it was not in cluster 2, so this cluster only retained two genes (BnaA06.GH3-19.a and BnaA06.GH3-8.d). The loss of *GH3* tandem repeat genes showed the changes in gene sequence and location after allopolyploidization. In *B. napus*, tandem duplication resulted in the generation of a new gene, namely BnaA09.GH3-8.g. The cluster of tandem repeat genes in *B. oleracea* was not retained in *B. napus*. A new cluster of tandem repeat genes appeared in *B. napus*, namely cluster 5, in which BnaC02.GH3-8.c and BnaC02.GH3-8.d were newly generated genes.

### 3.4. Phylogenetic Relationship of *GH3* Genes in *Arabidopsis* and *Brassica*

To explore the evolutionary relationships of the *GH3* genes from the three species of *Brassica* and *A. thaliana*, a phylogenetic tree was generated with 19 *A. thaliana*, 38 *B. rapa*, 25 *B. oleracea*, and 66 *B. napus* members, which would contribute to understanding the potential roles of the *GH3* genes in *Brassica* species. The phylogenetic tree was divided into three groups (I, II, and III), and this pattern of three groups for the *GH3* genes in the phylogenetic tree was in line with previous reports [23,24,46]. Group III possessed the highest number of the GH3 members (Figure 3).

A sister pair demonstrated the closest genetic relationship in a phylogenetic tree. A total of 54 sister pairs were found, including 7, 18, and 29 pairs in group I, II, and III, respectively. The overwhelming majority of the sister pairs were orthologous *GH3* gene pairs between A_n_/C_n_ and their respective ancestor genomes, with 31 A_n_−A_r_ pairs and 17 C_n_-C_o_ pairs. In group III, *AtGH3-18* and *AtGH3-19* were clustered together to become a sister pair, which suggested that the evolutionary relationship between *AtGH3-18* and *AtGH3-19* was closer than that between *AtGH3-18* and *AtGH3-19* and their respective *Brassica* orthologs. A similar scenario was observed for *AtGH3-14* and *AtGH3-15*. Some *AtGH3/BnaGH3* genes showed 1: n orthologous relationship, such as *AtGH3-8*/BnaA03.GH3-8.a, BnaAX.GH3-8.h, BnaC02.GH3-8.d, BnaC02.GH3-8.c, and BnaC02.GH3-8.b, and such an orthologous relationship proved that occurrence of *Brassica* genome triplication and allopolyploidization. All pairs of *BraGH3/BnaAGH3* and *BolGH3/BnaCGH3* showed a 1:1 relationship, which illustrated well conserved functions. No n:n relationships existed in *BraGH3/BnaAGH3* and *BolGH3/BnaCGH3*, suggesting no functional diversification in the three species of *Brassica*.

### 3.5. Gene Structure Analysis and Predicted Protein Characteristics

The 66 *BnaGH3* genes ranged from 1578 to 9498 bp in ORF (open reading frame) length, with an average of 2992 bp. These genes ranged from 1314 to 2379 in CDS (coding domain sequence) length, with an average of 1744 bp (Appendix A). Thus, the change in the intron length was broader than that in flanking exon length. To examine the differences in *GH3* ORF and CDS length between *B. napus* and the diploid ancestors, the ORF and CDS length of *GH3* genes in the diploid ancestors were analyzed (Appendix A). The 38 *BraGH3* genes ranged from 1888 to 4242 bp in ORF length, with an average of 1754 bp. The *BraGH3* genes ranged from 1530 to 1839 in CDS length, with an average of 1574 bp. The ORF length of the 25 *BolGH3* genes ranged from 1965 to 5355 bp, with an average of 2651 bp. The CDS length of the *BolGH3* genes ranged from 1413 to 1974 bp, with an average of 1756 bp. These results revealed that the average ORF length of the *BnaGH3* genes had larger differences from that of the *GH3* genes in the diploid ancestors, while the average CDS length was similar to the diploid ancestors, suggesting that intron length variation of the *BnaGH3* genes was more extensive than that of the *GH3* genes in the diploid ancestors.

The comparison of the structural pattern of exon-intron composition between *B. napus* and the diploid ancestors was performed. The exon number of *BraGH3* genes was from two to six, with most genes (63.2%) having four exons (Figure 4a). The exon number of *BolGH3* genes was from two to between six and eight. The number of *BolGH3* genes with three exons and genes with four exons were both 9 (37%) (Figure 4b). In *B. napus*, the number of *GH3* gene exons was from two to seven, and the genes (42.4%) with four exons was the most prevalent (Figure 4c). A new exon-intron organization appeared in *B. napus*, and two *BnaGH3* genes had seven exons. The variation of exon number was similar, and the difference was the proportion of genes with the same exon number between *B. napus* and the diploid ancestors. Finally, the gene structure between orthologous gene pairs (33 A_n_-A_r_ pairs and 20 C_n_-C_o_ pairs) was compared (Figure 5). For 33 A_n_-A_r_ pairs, there were three patterns, namely, *BnaGH3* genes had the same exon number as *BraGH3* genes, *BnaGH3* genes had more exons than *BraGH3* genes, and *BnaGH3* genes had less exons than *BraGH3* genes (Appendix A). Twenty A_n_-A_r_ orthologous pairs (60.6%) had the same exon number, in which 15 pairs had the same CDS length, but they had different ORF lengths, suggesting that differences in intron length led to the differences in orthologous gene structure. The remaining 13 A_n_-A_r_ orthologous pairs had different exon numbers, in which 11 *BnaGH3* genes had more exons than *BraGH3* genes. Although a few *BnaGH3* genes had more exons than *BraGH3* genes, their ORF lengths were almost the same as that of *BraGH3* genes (Appendix A). This may be because a part of the exon of *BraGH3* genes became an intron after allopolyploidization, such as BnaA01.GH3-5.a/BraA01.GH3-5.a. In addition, there is another possibility, and a part of the intron of *BraGH3* genes became an exon after allopolyploidization, such as BnaA03.GH3-11.a/BraA03.GH3-11.a. Finally, we found that BnaA06.GH3-12.c had less exons than BraA06.GH3-12.c, and the CDS length of BnaA06.GH3-12.c was also shorter than that of BraA06.GH3-12.c, but the ORF length of BnaA06.GH3-12.c was 1.7 times longer than that of the BraA06.GH3-12.c gene. This is because that the increase in the intron length lengthened the length of the *BnaGH3* gene. For 20 C_n_-C_o_ pairs, there were two patterns, namely, *BnaGH3* genes had the same exon number as *BolGH3* genes, and *BnaGH3* genes had more exons than *BolGH3* genes. Fifteen pairs of 20 (75%) had the same exon number, and eleven of these fifteen C_n_-C_o_ pairs also had the same CDS length (Appendix A). Two pairs not only had the same CDS length, but also owned the same ORF length. In the remaining 5 C_n_-C_o_ pairs, 5 *BnaGH3* genes had more exons than *BolGH3* genes, but not all *BnaGH3* genes had longer CDS than *BolGH3* genes. 

BnaC03.GH3-15.a owned shorter CDS, however, the increase in intron length allowed its ORF to be longer than BolC03.GH3-15.a. In summary, most *GH3* orthologous gene pairs between *B. napus* and the diploid ancestors had an almost identical gene structure pattern with regard to exon number and CDS length. These results demonstrated that the *GH3* gene family may be relatively conservative with regard to exon-intron composition, possibly because of their important roles in plant growth and development.

The various physicochemical parameters of GH3 proteins, including protein length, molecular weight, isoelectric point (pI), grand average of hydropathicity (GRAVY), and instability index, were calculated using ExPASy (Appendix A). The BnaGH3 protein length ranged from 437 aa to 792 aa, with an average of 580 aa. The molecular weight varied from 49.00 KD to 88.45KD, with an average of 65.51 KD. Except BnaA10.GH3-14.a and BnaC09.GH3-14.a, which had pI > 7, the BnaGH3 proteins had relatively low pI (pI < 7). The GRAVY value of BnaA09.GH3-8.f and BnaA06.GH3-8.e was greater than zero, which showed that these two proteins are hydrophobic. The rest of BnaGH3 proteins were hydrophilic, with a GRAVY value < 0. Forty-four (66.7%) BnaGH3 proteins possessed an instability index > 40, indicating that these proteins may be unstable. The physicochemical parameters of GH3 proteins in diploid ancestors were also calculated (Appendix A). The BraGH3 protein length ranged from 509 aa to 612 aa, with an average of 584 aa, and the BolGH3 protein length ranged from 470 aa to 657 aa, with an average of 584 aa. The average molecular weight of BraGH3 proteins and BolGH3 proteins were 65.84 KD and 65.99 KD, respectively. Only BraA10.GH3-14.a and BolC09.GH3-14.a had pI > 7, and their *B. napus* orthologous genes were BnaA10.GH3-14.a and BnaC09.GH3-14.a, respectively. Only BraA06.GH3-8.f had GRAVY > 0, and its *B. napus* orthologous gene was BnaA06.GH3-8.e. Twenty-five BraGH3 proteins (65.8%) and eighteen BolGH3 proteins (72%) owned an instability index exceeding 40. In a word, there was little difference between BnaGH3 proteins and GH3 proteins of two ancestors with respect to the average protein length and molecular weight. Two BnaGH3 proteins that had pI > 7 were derived from two diploid ancestors. One of two hydrophobic BnaGH3 proteins originated from *B. rapa*. The proportion of unstable proteins was the highest in *B. oleracea*, and the ratio in *B. napus* was similar to that in *B. rapa*.

Multiple amino acid alignment of 148 GH3 proteins (19 *A. thaliana*, 38 *B. rapa*, 25 *B. oleracea,* and 66 *B. napus* members) showed many relatively conserved sites and sequences, such as ^26^E and ^82^PYIDRI^87^ (Appendix A). There are seven highly conserved sites (^255^G, ^336^Y, ^340^E, ^346^N, ^362^P, ^380^L, ^386^G) and one highly conserved sequence (^366^YFEF^369^), which could be required for protein function. Chang et al. reported that the conserved motifs were essential to the adenylation activity for acyl-adenylate/thioester-forming enzyme superfamily [47]. The sequence conservation could correspond to functional conservation of GH3 proteins.

The availability of crystal structures of four *Arabidopsis* GH3 proteins (AtGH3-5, AtGH3-11, AtGH3-12, and AtGH3-15) helped us to compare the differences in secondary structure elements between the corresponding orthologous proteins. A total of 17 pairs of orthologous proteins were used for comparative analysis. Only one pair (BnaC09.GH3-12.c/BolC09.GH3-12.b) had the same number and location of α-helices and β-strands, suggesting that the secondary structure elements of most GH3 proteins changed after allopolyploidization (Appendix A). Two pairs (BnaC04.GH3-11.a/BolC04.GH3-11.b and BnaC03.GH3-15.a/BolC03.GH3-15.a) had larger differences in the protein secondary structure (Figure 6). BnaC04.GH3-11.a had six more α-helices and one more β-strand than BolC04.GH3-11.b, while BnaC03.GH3-15.a had four less α-helices and two less β-strands than BolC03.GH3-15.a. We found that BnaC04.GH3-11.a/BolC04.GH3-11.b and BnaC03.GH3-15.a/BolC03.GH3-15.a had different exon-intron composition, and the CDS length of these two pairs differed by several hundred bp, which may lead to larger differences in protein secondary structure. The protein 3-dimensional structures of the other pairs were shown in Appendix A.

### 3.6. Differences in Cis-Acting Elements of Promoter Regions

Cis-acting elements that are bound by transcription factors are associated with plant growth, development, and stress response [48]. Some phytohormone-related and stress-related cis-acting elements have been found in the promoters of *GH3* genes, including W box, G-box, CG box, AuxRE, and SARE [17,23]. To understand the potential functions of three *Brassica GH3* genes, phytohormone-related and stress-related cis-acting elements of the *GH3* promoter regions were screened out using PLACE. Seventy-eight, eighty, and eighty-five types of phytohormone-related and stress-related cis-acting elements were discovered in the promoters of *BraGH3*, *BolGH3*, and *BnaGH3* genes, respectively (Appendix A). *BraGH3* genes had one specific hypersensitive response element (S000466) and one specific auxin response element (S000368). *BolGH3* genes had one specific wounding response element (S000495) and two specific ABA response elements (S000015 and S000145), while *BnaGH3* gens had two specific G-box elements (S000345 and S000224) and one specific ABA response element (S000500). These results showed that the differences in cis-acting element types of *GH3* genes was very small among the three species of *Brassica*.

All *BraGH3* and *BolGH3* genes contained six common cis-acting elements, while all *BnaGH3* genes only had two common cis-acting elements, suggesting that changes in the promoter sequences of *BnaGH3* genes led to the loss of common cis-acting elements after allopolyploidization. There were 12 types of auxin response elements in ancestor species, and only one type (S000368) lost in *BnaGH3* genes, which indicated the types of auxin response elements had no significant differences between ancestor species and *B. napus*. The types of cis-acting elements were compared between orthologous gene pairs. For A_n_-A_r_ orthologous gene pairs, nine of thirty-three pairs had the same type of cis-acting elements (Appendix A). For five of these nine pairs, we found that the number of each type was also the same. After allopolyploidization, the promoter regions of these five *BnaGH3* genes retained the ancestral types of phytohormone-related and stress-related cis-acting elements and the number of each type, showing highly conservative behavior. The remaining orthologous gene pairs owned the different types of cis-acting element. Compared to corresponding *B. rapa* orthologous genes, some *BnaGH3* genes had specific types of cis-acting elements, such as BnaA03.GH3-5.b. Only two *BnaGH3* had no specific type. For C_n_-C_o_ orthologous gene pairs, only one pair had the same type of cis-acting element, and the number of each type was also the same. The rest of the orthologous gene pairs owned the different types of cis-acting element (Appendix A). Seventeen *BnaGH3* genes had specific types of cis-acting elements relative to corresponding *B. oleracea* orthologous genes, such as BnaC04.GH3-11.a. Similarly, there are only two *BnaGH3* genes with no specific type. To sum up, most *BnaGH3* genes had specific cis-acting element types relative to orthologous diploid genes, and the promoter sequence changes of *BnaGH3* genes on the C_n_ subgenome was slightly larger than that of *BnaGH3* genes on the A_n_ subgenome. Although obvious differences in cis-acting element types between orthologous gene pairs existed, all types of cis-acting elements in *B. napus* had no obvious difference compared with the ancestor species, suggesting that the *GH3* gene family remained phytohormone-related and stress-related cis-acting elements from diploid ancestors after allopolyploidization.

### 3.7. Expression Patterns of *GH3* Genes in the Three Species of *Brassica*

To understand the expression changes of *GH3* genes, we analyzed *GH3* gene expression patterns across four different tissues (stem, leaf, flower, and silique) in the three species of *Brassica*, and compared the expression pattern of *GH3* genes between duplicated genes based on RNA-seq data (Appendix A). The *GH3* gene expression patterns in four different tissues for each *Brassica* species were shown in Figure 7. Six *BraGH3* genes, four *BolGH3* genes, and five *BnaGH3* genes had no expression in four tissues, of which eleven were *GH3-8* genes (4 *BraGH3-8* genes, 3 *BolGH3-8* genes, and 4 *BraGH3-8* genes). The number of *GH3-8* genes was the highest in all three species. To avoid functional redundancy, some *GH3-8* genes may be non-functional or inducible. Compared with ancestor species, *B. napus* had more tissue-specific expressed genes, suggesting that allopolyploidization increased the spatial specificity of gene expression.

To understand how the expression of ancestral genes changed after allopolyploidization, the expression patterns of the orthologous gene pairs were compared between A_n_ and A_r_, C_n_, and C_o_. The results demonstrated that the vast majority of the orthologous gene pairs had disparate expression patterns across four tissues, which reflected gene expression changes after allopolyploidization (Figure 8). Some genes lacked expression in all four tissues in ancestors, however, their orthologous genes were expressed at low levels in *B. napus*, such as BraA06.GH3-8.e/BnaA06.GH3-8.d. In addition, a few genes in *B. rapa* had lower expression levels, while their orthologous genes in *B. napus* were not expressed in all four tissues, such as BraA03.GH3-8.b/BnaA03.GH3-8.a. Finally, some ancestral genes showed high expression levels in tissues, whereas their orthologous genes in *B. napus* showed low expression levels. For example, BolC07.GH3-5.b had higher expression levels in leaf, while BnaC07.GH3-5.b had the lowest expression level in leaf. On the contrary, some ancestral genes showed low expression levels in tissues, whereas their orthologous genes in *B. napus* showed high expression levels. For example, BraA06.GH3-7.a was expressed at the lowest level in silique, while BnaA06.GH3-7.a was expressed at the highest level in silique.

Zhou et al. reported that some duplicated genes of *bZIP* transcription factor family in *B. napus* had opposite expression patterns in seventeen tissues [49]. To comprehend whether the *GH3* gene family in *B. napus* also had this phenomenon, we compared the expression patterns of the duplicated genes. The duplicated genes had a diverged expression pattern. Tandem duplication gene BnaA06.GH3-7.a and BnaA06.GH3-12.c showed opposite expression patterns. BnaA06.GH3-12.c lacked expression in all tissues, however, BnaA06.GH3-7.a had high expression in all tissues. The expression patterns of the most homologous gene pairs between A_n_ and C_n_ were diverse. For example, BnaA03.GH3-12.b had lower expression in all tissues (FPKM < 1 ), and the expression levels in the leaf and the silique were comparable. Its duplication gene BnaC03.GH3-12.b had higher expression in all tissues, and the expression level in the silique is 1.8 times higher than that of the leaf. In addition, the tissue-specific expression of the duplicated genes was also different. BnaC02.GH3-16.a was the flower-specific expression, while BnaAX.GH3-16.a was the silique-specific expression. Finally, a few duplicated gene pairs shared similar expression patterns in four tissues, such as BnaA03.GH3-11.a and BnaCX.GH3-11.b.

In summary, the expression levels of the ancestral species *GH3* genes changed obviously after allopolyploidization. Duplicated gene pairs in *B. napus* showed diverse expression patterns, which could be due to the functional divergence during the process of evolution.

## 4. Discussion

In recent years, increasing studies have showed that *GH3* genes played an important role in plant growth and development, but the genome-wide comprehensive analysis of the *GH3* gene family in *B. napus*, and how the exon-intron organization, cis-element distribution, and gene expression level of the *BnaGH3* genes change after allopolyploidization still remain largely unknown. In this study, we confirmed the origin of *BnaGH3* genes and analyzed changes in *BnaGH3* gene exon-intron composition, cis-element distribution, and expression pattern relative to diploid ancestors. These results may provide a reference for exploring the molecular mechanisms of the adaptive enhancement in *Brassica* allopolyploid.

### 4.1. The *Brassica* Genome Triplication and Allopolyploidization Bring about the Expansion and Loss of *GH3* Genes

The whole genome duplication (*Brassica* WGT and allopolyploidization) contributes the most to the expansion of gene families in *B. napus*, such as the *Aux/IAA* gene family, the *BES1*gene family, and the *AP2/ERF* gene family [44,45,50]. Our results showed that the expansion of the *GH3* gene family was also mainly affected by the whole genome duplication. Previous research reported that whole genome duplication had a significant effect on the amplification of the *GH3* gene family in legumes and apples [46,51].

The *Brassica* genus underwent an extra whole genome triplication relative to Arabidopsis, therefore, one *A. thaliana* gene should possess three corresponding *B. rapa* or *B. oleracea* orthologous genes and six *B. napus* orthologous genes [52]. With 19 *GH3* genes in *A. thaliana*, there should be theoretically about 60 and 120 *GH3* genes in *B. rapa/B. oleracea* and *B. napus*. However, only 38, 25, and 66 *GH3* genes were found in *B. rapa*, *B. oleracea*, and *B. napus*, respectively. The number of *GH3* genes in the A_n_ and C_n_ was 39 and 27 in *B. napus*, which was almost the same as that in A_r_ (*B. rapa*) and C_o_ (*B. oleracea*) genomes. These results demonstrated that the loss of most *GH3* genes happened after *Brassica* genome triplication. For example, *AtGH3-1* had only one corresponding *B. rapa* and *B. oleracea* orthologous gene, so only two *GH3-1* genes remained in *B. napus*. *AtGH3-4* had no corresponding *B. rapa* and *B. oleracea* orthologous gene, which could lead to the absence of *GH3-4* in *B. napus*. After *Brassica* genome triplication, 35% genes were lost in the *Brassica* lineage, most likely through a deletion mechanism [53]. The *Brassica* WGT resulted in more *GH3* genes in the *Brassica* diploid progenitors than in *A. thaliana*, which was followed by gene loss. Appendix A showed the time of WGT and allopolyploidization events and the number of *GH3* genes that were missing or new in the different times.

We found that some *GH3* genes in diploid ancestors were lost after allopolyploidization. Five *BraGH3* genes and five *BolGH3* genes had no corresponding *B. napus* orthologous genes, which suggested that these genes could be lost during the formation of allotetraploid. There were three possibilities for the absence of ancestor *GH3* genes. The first possibility, *B. napus* went through rapid inter-chromosomal rearrangements and chromosomal fragment losses following allopolyploidization, which resulted in these *GH3* genes’ losses from the *B. napus* genome [54]. The second possibility, these *GH3* genes became pseudogenes via accumulating deleterious mutants, and pseudogenes could be deleted or diverge too extensively to be confirmed during the evolution process of *B. napus* [55]. The final possibility, the absence or incompleteness of the GH3 functional domain caused these ancestor GH3 proteins to no longer be GH3 proteins after allopolyploidization, which could be due to the changes of ancestor *GH3* gene structures or sequences. The gene loss has been regarded as an advantageous event during the genome evolution [56]. The loss of a gene encoding an anthocyanin pathway enzyme was involved in the transition from blue to red flowers, which led to phenotypic differences among *Andean Iochroma* species [57]. The deletion of a penicillin-binding protein was associated with the resistance to the cephalosporin drug ceftazidime [58]. The loss of these ancestor *GH3* genes may play beneficial and adaptive roles in the evolution process of *B. napus*.

Several molecular mechanisms are involved in the production of new genes, such as gene duplication, exon shuffling, lateral gene transfer, and so on [59]. In our results, thirteen new *BnaGH3* genes appeared after allopolyploidization. Changes in the gene structures or sequences of seven ancestor *GH3* genes (their corresponding proteins had incomplete functional domains) resulted in the reacquisition of the complete GH3 domains during the allopolyploidization. For example, Bra004432 protein had an incomplete GH3 domain, so it was not considered as a GH3 protein, while its orthologous protein BnaA05.GH3-9.a had a complete GH3 domain after the formation of *B. napus*. The tandem duplication was one of the expansion mechanisms of the *GH3* gene family in *Medicago truncatula* [17]. We also found that three new *BnaGH3* genes originated from the tandem duplication. The production of the remaining three new *BnaGH3* genes may be due to the following molecular mechanisms. The first molecular mechanism was the retroposition. MRNAs are reverse-transcribed into double strand cDNAs and then incorporated back into new genomic positions to form retrogenes. Many retrogenes have been identified in rice, and most of them go through negative selection, suggesting that these retrogenes are likely functional [60]. The second molecular mechanism was the lateral gene transfer. The exchanges of genes between different species, organelles, and nuclei can create new genes. Mitochondrial genes that encode respiratory and ribosomal proteins are subject to lateral transfer between flowering plants [61]. The final molecular mechanism was the exon shuffling. Exons from different genes are reassembled to form a novel exon–intron structure. A defensin gene was inserted into an exon that was from a gene belonging to another subfamily, and the tissue specificity expression of this gene was still preserved [62]. Since new genes play an import role in plant growth, development, and defense [63,64,65], the appearance of new *BnaGH3* genes may contribute to the adaptability improvement of *B. napus*.

### 4.2. Variant Expression Patterns of *BnaGH3* Genes May Contribute to the Adaptability Improvement in *Brassica napus*

Gene expression changes in allopolyploids could be an adaptive mechanism, which makes allopolyploids evolve rapidly and steadily in nature [66]. The expression changes in genes mediated by circadian clock genes resulted in strong growth vigor and increased biomass in Arabidopsis allotetraploids [67]. Compared with *Brassica* diploids, *Brassica* allopolyploids showed different gene expression patterns, which could contribute to survival and evolution [68,69]. In addition, the members of the gene families also exhibited distinct expression patterns across different tissues in *Brassica* allotetraploid, suggesting that some genes may be sub-functionalized or neo-functionalized during the process of the evolution [44,70,71].

After allopolyploidization, the expression levels of some *GH3* genes showed significant changes. Besides, the expression patterns of most orthologous pairs are different. Compared with diploid ancestors, the expression levels of a few *GH3* genes were greatly enhanced in *B. napus*. For example, BraA06.GH3-7.a had the lower expression levels (FPKM < 1.2) in four tissues, while the FPKM of its orthologous BnaA06.GH3-7.a ranged from 24.21 to 83.66 in four tissues. It is reported that the *GH3-7* gene could play important roles in drought and salt stress [46,51]. The high expression of BnaA06.GH3-7.a may increase the stress tolerance of *B. napus*. In addition, Zeng et al. reported that *GH3-7* had important functions in peach fruit ripening [72]. The expression level of BnaA06.GH3-7.a was the highest in the silique, showing that it could play a vital role in controlling the silique development in *B. napus*. The *GH3-12* gene participated in salicylic acid metabolism and activated defense responses in Arabidopsis leaves [73]. BolC03.GH3-12.a had FPKM < 1 in leaf, while its orthologous BnaC03.GH3-12.b had a relatively high level (FPKM = 4.9) in leaf, suggesting that the increased expression level of BnaC03.GH3-12.b may improve the defense ability in *B. napus* leaf. Relative to diploid ancestors, the expression levels of a few *GH3* genes in *B. napus* were greatly decreased. For example, the FPKM of BolC07.GH3-5.b was 63.77, while the FPKM of BnaC07.GH3-5.b was 1.49 in the stem. The FPKM of BolC03.GH3-15.a ranged from 54.42 to 87.51 across four tissues, while the FPKM of BnaC03.GH3-15.a varied from 6.05 to 9.96. The large drop in the expression levels may be due to the increase in the number of genes in *B. napus*, and measures were taken to avoid gene function redundancy.

The molecular mechanisms of gene expression changes in polyploidies have been explored. The first molecular mechanism is altered regulatory networks. Gene expression is controlled by regulatory networks composed of various interrelated regulatory factors, such as transcription factors, and altered regulatory networks could affect the phenotype and contribute to heterosis in polyploids [74]. The genetic changes are associated with alterations in DNA sequences, which also have effects on gene expression [74]. After the formation of *Brassica* allopolyploids, the gene structures had undergone rapid and extensive changes caused by DNA rearrangements, homoeologous recombination, gene conversion-like events, etc. [75]. Insertion of transposons into genes or locations near the genes could alter the gene expression pattern or gene structure [76]. Genome rearrangements significantly changed the gene expression level in *B. napus*, which could cause phenotypic variability [77]. The final molecular mechanism is epigenetic changes, such as DNA methylation, miRNA regulation, and so on. Alterations in DNA methylation patterns brought about transcriptional changes and phenotypic instability in *Arabidopsis* allotetraploids [78]. Ha et al. reported that changes in miRNA expression could give rise to variation in gene expression and adaptation [79]. In our results, genetic structure changes may be one of the reasons for changes in *BnaGH3* gene expression. The exon-intron organization of BnaA05GH3-18.a was different from BraAX.GH3-19.e after allopolyploidization, which may lead to d_N_/d_S_ >> 1. BnaA05GH3-18.a was subjected to strong positive selection, so it had a distinct expression pattern. BnaA03.GH3-8.a and BnaAX.GH3-8.h may be also like this. In addition, gene expression could be regulated by cis-elements in response to various environmental stress or developmental programs [80]. We found that the majority of orthologous gene pairs between *B. napus* and ancestor species had differences in the types and number of cis-elements, which might lead to variation in *BnaGH3* gene expression.

A direct result of allopolyploidization is gene duplication. There are several cases for duplicated genes in allopolyploids: (1) duplicated genes retain original function; (2) one of duplicated genes is neo-functionalized; (3) one of the duplicated genes is silenced; (4) duplicated genes are sub-functionalized [81,82]. The study on homologous genes derived from three diploid species in allohexaploid wheat indicated that homologous genes not only had similar expression patterns, but also had distinct expression patterns in different tissues [83]. The functional divergence of duplicated genes is a main trait during the long-term evolution process for polyploids [84]. In our results, a few homologous gene pairs had similar expression patterns. BnaA03.GH3-11.a and BnaCX.GH3-11.b had the highest expression level in the flower, followed by the silique, stem, and leaf, indicating that they could play an important role in the flower development. Many homologous gene pairs had different expression patterns. For example, BnaA06.GH3-7.a had higher expression level across four tissues, while BnaC06.GH3-7.a was barely expressed across the four tissues, showing the emergence of functional divergence. This differential expression between homologous genes in polyploids may have a profound effect on plant evolution. On the one hand, the functional divergence of homologous genes could protect redundant genes from being eliminated in natural selection during long-term evolution. On the other hand, the distinct expression patterns of homologous genes in different tissues may also lead to the production of new phenotypes, promoting plant diversity.

## Figures and Tables

**Figure 1 genes-10-00058-f001:**
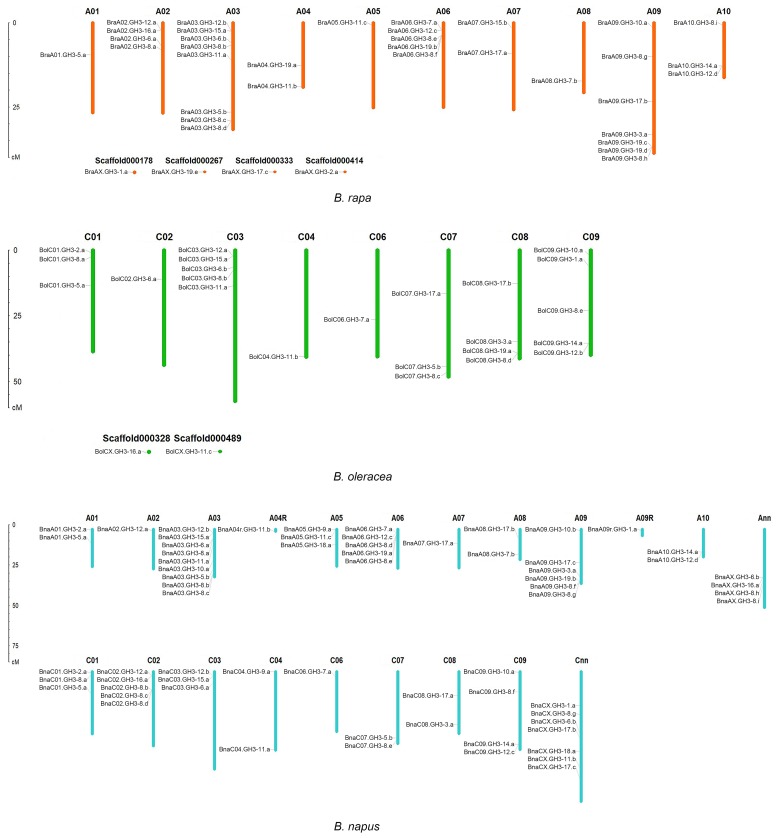
The chromosomal location of *GH3* genes in the three species of *Brassica*. Scaffords: assembled sequence that failed to be incorporated into the chromosomes; A04R means A04-random; A09R means A09-random; A04-random and A09-random: the specific positions of genes on A04 and A09 are still unknown; A_nn_ means A_nn_-random; C_nn_ means C_nn_-random; A_nn_-random and C_nn_-random: unmapped A chromosomes of *B. napus* genome and unmapped C chromosomes of *B. napus* genome.

**Figure 2 genes-10-00058-f002:**
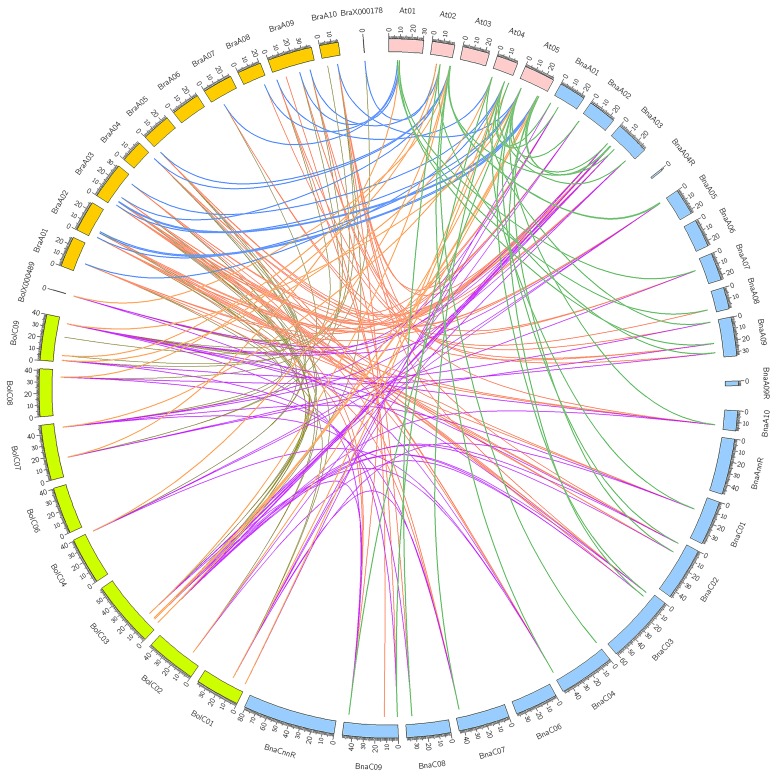
The synteny analysis of *GH3* genes in the three species of *Brassica* and *Arabidopsis thaliana* chromosomes.

**Figure 3 genes-10-00058-f003:**
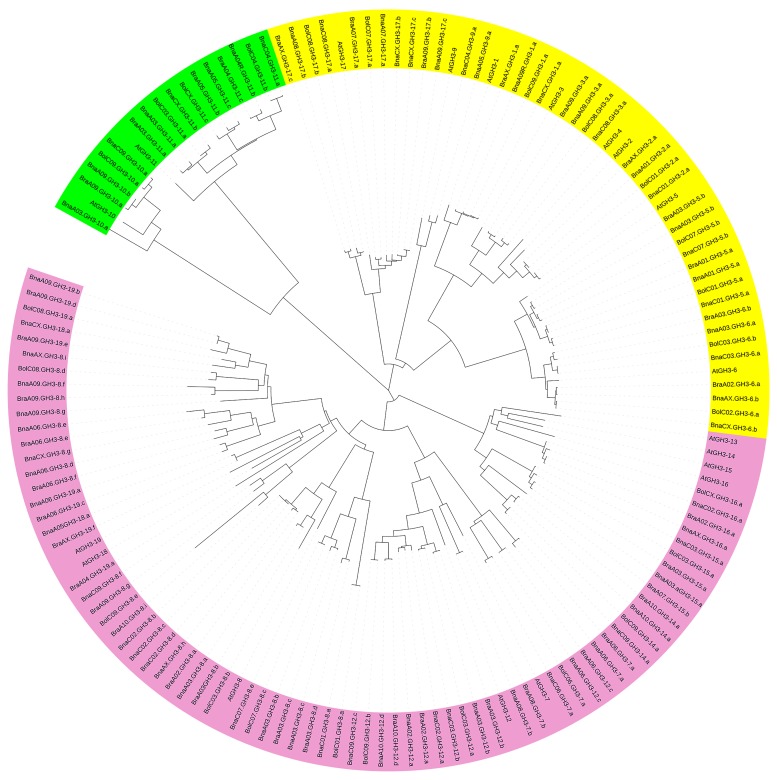
The phylogenetic relationship of *GH3* genes among *B. rapa*, *B. oleracea*, *B. napus*, and *A. thaliana*. The green, yellow, and pink regions represent group I, group II, and group III, respectively.

**Figure 4 genes-10-00058-f004:**
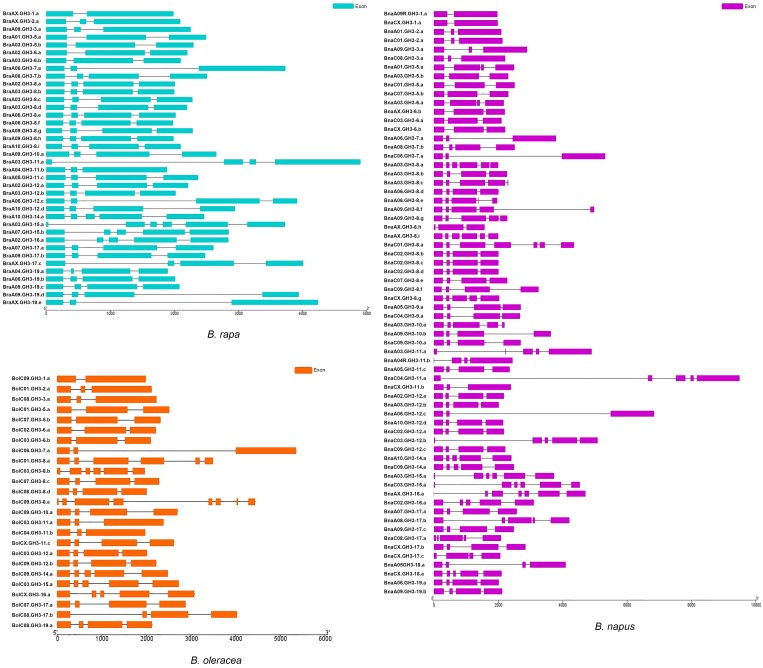
The exon–intron compositions of *GH3* genes in the three species of *Brassica*.

**Figure 5 genes-10-00058-f005:**
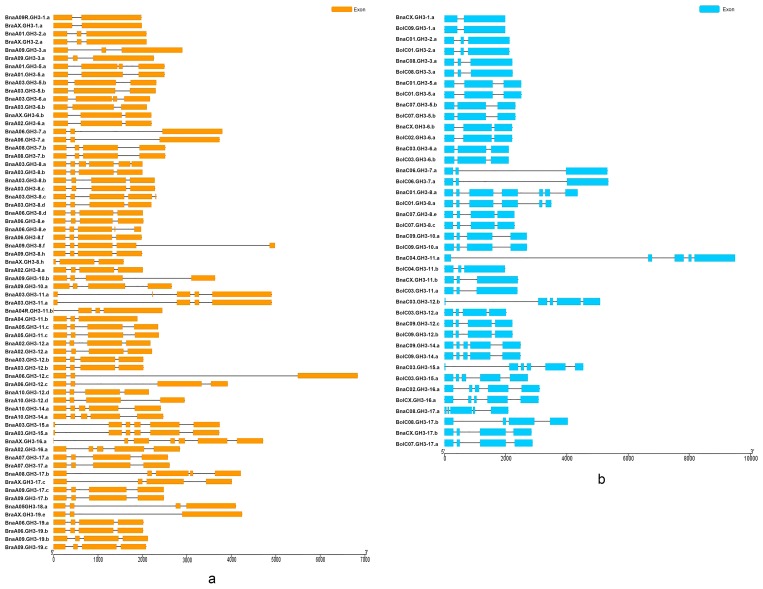
The exon–intron compositions of 33 *GH3* A_n_-A_r_ orthologous pairs and 20 *GH3* C_n_-C_o_ orthologous pairs. (**a**) Thirty-three *GH3* A_n_-A_r_ orthologous pairs; (**b**) twenty *GH3* C_n_-C_o_ orthologous pairs.

**Figure 6 genes-10-00058-f006:**
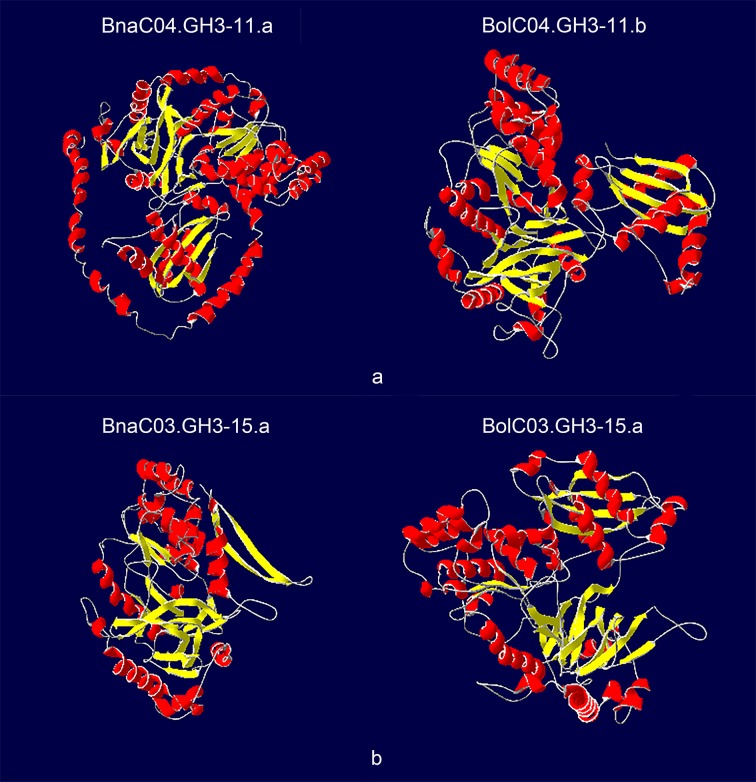
The 3-dimensional structures of BnaC04.GH3-11.a/BolC04.GH3-11.b and BnaC03.GH3-15.a/BolC03.GH3-15.a. The α-helices, β-strands, and random coils were colored by red, yellow, and grey, respectively.

**Figure 7 genes-10-00058-f007:**
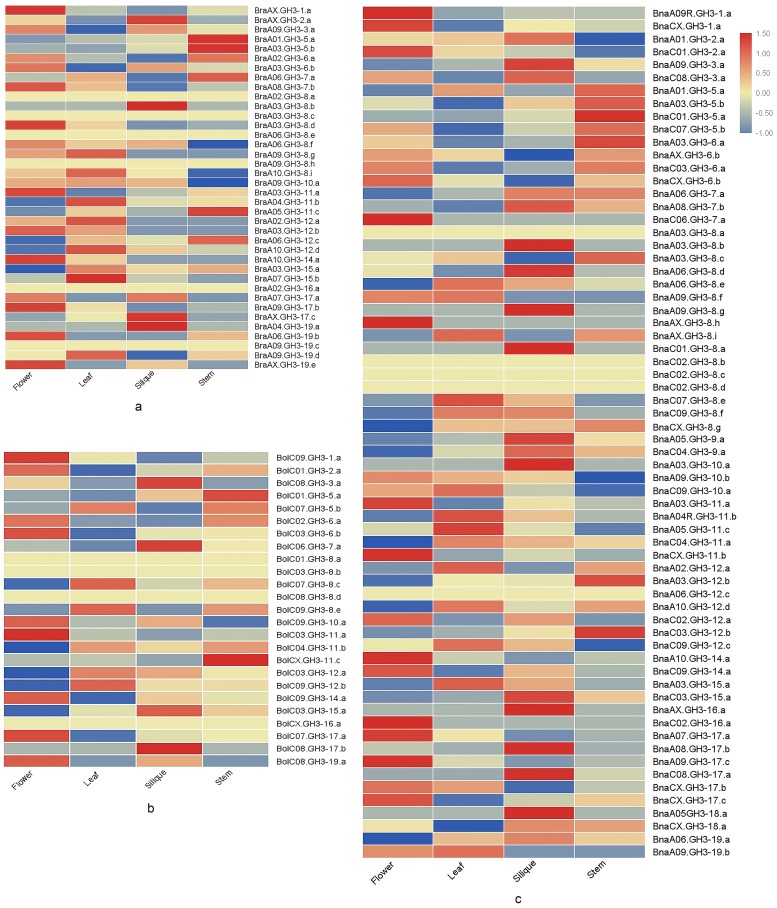
The expression profiles of *GH3* genes in different tissues for the three species of *Brassica*. (**a**) *B. rapa*; (**b**) *B. oleracea*; (**c**) *B. napus*.

**Figure 8 genes-10-00058-f008:**
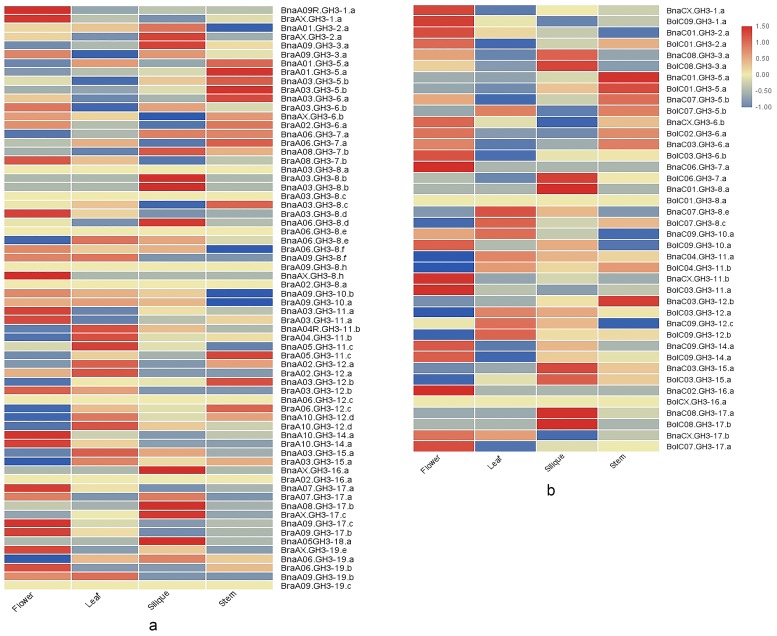
The comparison of expression patterns between orthologous pairs. (**a**) Thirty-three *GH3* A_n_-A_r_ orthologous pairs; (**b**) twenty *GH3* C_n_-C_o_ orthologous pairs.

**Table 1 genes-10-00058-t001:** The number of *GH3* genes in the three species of *Brassica*.

*GH3* Gene	*Brassica* *rapa*	*Brassica* *oleracea*	*Brassica* *napus*
*GH3-1*	1	1	2
*GH3-2*	1	1	2
*GH3-3*	1	1	2
*GH3-4*	0	0	0
*GH3-5*	2	2	4
*GH3-6*	2	2	4
*GH3-7*	2	1	3
*GH3-8*	9	5	16
*GH3-9*	0	0	2
*GH3-10*	1	1	3
*GH3-11*	3	3	5
*GH3-12*	4	2	7
*GH3-13*	0	0	0
*GH3-14*	1	1	2
*GH3-15*	2	1	2
*GH3-16*	1	1	2
*GH3-17*	3	2	6
*GH3-18*	0	0	2
*GH3-19*	5	1	2
Total	38	25	66

**Table 2 genes-10-00058-t002:** The number of *GH3* genes on each chromosome.

The Number of *GH3* Gene	*B. rapa* Chromosome	*B. oleracea* Chromosome	*B. napus* Chromosome
1	A_r_01, A_r_05, A_r_08	C_o_02, C_o_04, C_o_06	A_n_02, A_n_04-random, A_n_07, A_n_09-random, C_n_06
2	A_r_04, A_r_07	Scaffords	A_n_01, A_n_08, A_n_10, C_n_04, C_n_07, C_n_08
3	A_r_10	C_o_01, C_o_07	A_n_05, C_n_01, C_n_03
4	A_r_02, Scaffords	C_o_08	A_nn_-random, C_n_09
5	A_r_06	C_o_03, C_o_09	A_n_06, C_n_02
6			A_n_09
7	A_r_09		C_nn_-random
8	A_r_03		
9			A_n_03

Scaffords: assembled sequence that failed to be incorporated into the chromosomes; A_nn_-random: unmapped A chromosomes of *B. napus* genome; C_nn_-random: unmapped C chromosomes of *B. napus* genome; A_n_04-random and A_n_09-random: the specific positions of genes on A_n_04 and A_n_09 are still unknown.

**Table 3 genes-10-00058-t003:** The clusters of GH3 tandem repeat genes in the three species of *Brassica*.

Cluster	*B. rapa*	*B. oleracea*	*B. napus*
Cluster 1	BraA06.GH3-12c^a^	BolC08.GH3-8.d	BnaA06.GH3-12.c ^a^
	BraA06.GH3-7.a^b^	BolC08.GH3-19.a	BnaA06.GH3-7.a ^b^
Cluster 2	BraA06.GH3-8.f		BnaA06.GH3-19.a ^c^
	BraA06.GH3-19.b^c^		BnaA06.GH3-8.d ^d^
	BraA06.GH3-8.e^d^		
Cluster 3	BraA09.GH3-8.h^e^		BnaA09.GH3-8.f ^e^
	BraA09.GH3-19.d		BnaA09.GH3-19.b ^f^
	BraA09.GH3-19.c^f^		BnaA09.GH3-8.g
Cluster 4	BraA03.GH3-8.d^g^		BnaA03.GH3-8.c ^g^
	BraA03.GH3-8.c^h^		BnaA03.GH3-8.b ^h^
Cluster 5			BnaC02.GH3-8.b
			BnaC02.GH3-8.c
			BnaC02.GH3-8.d

Genes with the same superscript letter belong to the same set of orthologous gene. The letters “a” to “h” represent orthologous gene pairs.

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
