# Peer review of "The Gene Structure and Expression Level Changes of the GH3 Gene Family in Brassica napus Relative to Its Diploid Ancestors"

_genes, 2019, doi:10.3390/genes10010058_

Round 1
Reviewer 1 Report
In this manuscript, the authors studied GH3 genes in Brassica napus and compared them to Brassica rapa and Brassica oleracea with respect to gene structures, expression levels and cis-regulatory elements. This study is very extensive and interesting. I have the following suggestions and questions:
We learn about An and Cn in the introduction, however they are also used in the abstract in line 17. It might be difficult to understand the abstract for readers who are not familiar with subgenome.
In line 33, please define allopolyploidy.
In line 45, what does this mean : "one gene was specifically activated" ?
To what degree the gene loss and gains could be due to mis-annotation?
Lines 176-196 are hard to follow
Are the heatmaps in Figures 7-8 sorted or clustered in any way? Also the key in this figure is missing. Are the values fold change values and what is being compared?
Overall some parts of the manuscript could also be shortened by summarizing the major findings instead of listing all genes, which could be given in supplemental data.
Author Response
Reviewer #1:
Comments and Suggestions for Authors
In this manuscript, the authors studied GH3 genes in Brassica napus and compared them to Brassica rapa and Brassica oleracea with respect to gene structures, expression levels and cis-regulatory elements. This study is very extensive and interesting. I have the following suggestions and questions:
1. We learn about An and Cn in the introduction, however they are also used in the abstract in line 17. It might be difficult to understand the abstract for readers who are not familiar with subgenome.
Response:
Thank you very much for your valuable comments. The genomic information was added in line 16 of the revised manuscript. The subgenome is a description of the polyploid genome, and multiple subgenomes make up the polyploid genome.
2. In line 33, please define allopolyploidy.
Response:
Thank you very much for your valuable suggestions. The definition of allopolyploid was added in lines 33-35 of the revised manuscript.
3. In line 45, what does this mean: "one gene was specifically activated" ?
Response:
Thank you very much for your helpful comments. The researcher explained that the transcripts of AFLP-10 gene accumulated at a very low level in the two parental lines but were abundant in the synthetic hexaploid plants, so this gene was specifically activated in the synthetic hexaploid plants
4. To what degree the gene loss and gains could be due to mis-annotation?
Response:
Thank you very much for your careful review. The protein sequences of the three species were used to search the GH3 genes rather than gene annotation information, so mis-annotation information did not affect the discovery of the GH3 genes. The genome annotation files mentioned in line 106 referred to the GFF files, which contained the gene structure and location information (number and location of exon and intron). GFF files were used to make the gene structure map. 19 GH3 protein sequences of A. thaliana and the Hidden Markov Model (HMM) profile of GH3 domain were used to search all protein sequences of each Brassica species. If the searched protein had a complete GH3 functional domain, the gene that the protein corresponded to was a GH3 gene.
5. Lines 176-196 are hard to follow
Response:
Thank you very much for your helpful comments. We counted the number of GH3 genes on each chromosome to compare the gene number distribution between An and Ar, Cn and Co. The content in lines 176-179 of the previous manuscript has been rearranged in lines 198-211 of the revised manuscript, and a new table was added, named Table 2.
6. Are the heatmaps in Figures 7-8 sorted or clustered in any way? Also the key in this figure is missing. Are the values fold change values and what is being compared?
Response:
Thank you very much for your critical comments. This is my omission. The FPKM values of all GH3 genes were standardized and converted to Z-values. Z-values were used to make heatmaps by TBtools. The details were added to Materials and Methods in lines 171-174 of the revised manuscript. Figure 7 showed that expression levels of GH3 genes were compared among four tissues in each species, and the used values were Z-values. Figure 8 showed that expression patterns of GH3 genes in four tissues were compared between orthologous pairs, and the used values were also Z-values.
7. Overall some parts of the manuscript could also be shortened by summarizing the major findings instead of listing all genes, which could be given in supplemental data.
Response:
Thank you for your insightful suggestions. The content in lines 208-215 of the previous manuscript has been rewritten in lines 234-236 of the revised manuscript, an attached table was added and named Table S3. The sentence in lines 255-258 of the previous manuscript has been rewritten in lines 283-284 of the revised manuscript. The sentence in lines 405-408 of the previous manuscript has been rewritten in lines 427-429 of the revised manuscript. The content in lines 483-491 of the previous manuscript has been rewritten in lines 500-503 of the revised manuscript. The content in lines 510-512 in the previous manuscript was redundant, so it was deleted.
Reviewer 2 Report
The manuscript by Wang et al represents a detailed analysis of the GH3 genes in the 3 species B. rapa, B.oleraceae and B.napus. The authors have made use of available genome data to identify the GH3 genes in the three species and their chromosomal distribution, have analysed the orthologous relationship and phylogeny, looked at gene structure and predicted protein characteristics and finally compared the expression pattern in different tissues in the 3 species. The data collected could represent valuable information for researchers interested in the functional characterization of specific GH3 genes in B. napus. I nevertheless have some comments on the manuscript.
1. More descriptive figure legends could help readers not totally familiar with the subject or the plant species to more quickly understand the data. E.g in figure 1: what is indicating Cnn? Ann? A04R? , in Fig 3: which are group 1 group 2 and group3? Figure 4 and 5: what is the point of the different colour of the exons in the 3 species? Figure 7 and 8: what is the use of a, b and c panels? It is not indicated in the legend nor in the text.
2. A less descriptive text. In my opinion most of the information regarding the specific genes should be available via the figures and collected data that should be made available to the readers, rather than in the manuscript text. At the moment, the main take home messages result rather diluted within the details in the text and difficult to identify. The long text (with recurrent names of the genes) results a bit repetitive, difficult to read with the risk of losing interest.
3. English: some sentence do not make very much sense to me. Among others: lines 43-46; line 85: “understanding for the function” should be changed into “a better understanding of the function”; line 24: the comprehensive analysis. “the” should be changed into “a comprehensive analysis”.
4. Abstract: line 14: “the effect of allopolyplodization on GH3…”I am surprised that the authors only mentioned allopolyploidization while in their conclusions they state that most gene loss occurred after WGT. Was that finding already reported? If not I would rephrase as “the effect of WGT and of allopolyplodization on GH3…”
5. Introduction: Line 61-62. It is stated that group 3 is Arabidopsis specific but this can´t be true if it is later found at least also in the 3 brassicas analysed. Line 70-71: how about apple and legumes as cited in line 535?
6. Materials and methods: line 99 “Plants were grown in natural environment” is definitively too vague and should be described more into details especially with regards to the RNA seq analyses performed. In 2.8 there is no trace of how plants were grown, which material was harvested, how, when during development, at which time of the day etc..all variables that highly influence gene expression especially when comparisons want to be done. In 2.7, for the search of cis-elements, standard 2000 bp of upstream genomic seq were considered for all the species. However, research in this paper shows that intragenic and possibly intergenic regions were expanded in B. napus compared with ancestors. Perhaps the whole intergenic region until the upstream genes should be taken in consideration. Moreover, regulatory elements can also be present within introns and in intergenic regions downstream of the genes but this was not taken in consideration.
7. Result 3.1 when talking about the lack of copies (e.g. AtGH3-4 and -13) does this mean that there are also no pseudogenes? I think it would be valuable information to know whether the gene is there in the same synthenic position but it is not functional anymore (especially with regard to discussion line 552-560 and at the end of 3.2). Moreover, in the case of pseudogenes, I would not say that a genes disappear. I think this would be misleading.
8. Result 3.2 line 205-206: “most of the GH3 genes remain intact”. This chapter it is not about the gene but about the synthenic region, I guess. line 207: not sure where the 13 number came from (my calculation was 38 minus 33 from An-Ar and 25 minus 20 from Cn-Co, so 10?).
Lines 208-215: if I understood correctly it is suggested that an incomplete GH3 domain became a complete one after allopolyploidization? I would rather argue that the GH3 domain in B. rapa or B.oleraceae has become incomplete in more recent times than when the allopolyploidization took place, whereas it has been kept complete in B. napus.
Line 221-224: very unclear to me what the authors want to tell. Is it about this study? About common knowledge? Former literature? Than it should be cited and should be made it clearer. In the case of common knowledge, the use of the past tense “syntenic genes were..” would be wrong then. Present should be used. The same in line 233.
Line 228: “demonstrated”, I would say “confirmed”
9. Results 3.5 which data has been used for the gene structure analyses? Expression data already available? Was the structure confirmed using the RNA seq performed in this research?
10. Discussion 4.1 I find strange expression such as line 542 “ the loss of most GH3 genes occurred in the process of brassica genome triplication”. I would think that the loss happened somewhat after the WGT, not during or in the process. Or again line 548 “accompanied by gene loss” I would think “followed by gene loss”?. I would suggest the author to make a graphic representation of the timing of events WGT and allopolyploidization and corresponding genes that were lost or that arise in in the different time. It would help the reader to understand.
11. Discussion 4.1. I wonder how much variation in GH3 there is already at the species level rather than at the interspecific level among the tree species analysed here (which could lead to incorrect conclusions). In Arabidopsis several accessions have been sequenced. Is there variation in GH3 genes within the same species? Perhaps a disclaimer should be made on this point.
Line 534-535: it should make clearer that the report from the previous researcher was from other species.
Author Response
Reviewer #2:
Comments and Suggestions for Authors
The manuscript by Wang et al represents a detailed analysis of the GH3 genes in the 3 species B. rapa, B.oleraceae and B.napus. The authors have made use of available genome data to identify the GH3 genes in the three species and their chromosomal distribution, have analysed the orthologous relationship and phylogeny, looked at gene structure and predicted protein characteristics and finally compared the expression pattern in different tissues in the 3 species. The data collected could represent valuable information for researchers interested in the functional characterization of specific GH3 genes in B. napus. I nevertheless have some comments on the manuscript.
1. More descriptive figure legends could help readers not totally familiar with the subject or the plant species to more quickly understand the data. E.g in figure 1: what is indicating Cnn? Ann? A04R? , in Fig 3: which are group 1 group 2 and group3? Figure 4 and 5: what is the point of the different colour of the exons in the 3 species? Figure 7 and 8: what is the use of a, b and c panels? It is not indicated in the legend nor in the text.
Response:
Thank you very much for your valuable suggestions. This is my omission. The missing information was added in lines 218-222, line 331, lines 494-495 and lines 511-512 of the revised manuscript. In figure 4, three colors represent three species. Blue, yellow, and purple represent B. rapa, B. oleracea, and B. napus, respectively. In figure 5, yellow and blue represent 33 GH3 An-Ar orthologous pairs and 20 GH3 Cn-Co orthologous pairs, respectively.
2. A less descriptive text. In my opinion most of the information regarding the specific genes should be available via the figures and collected data that should be made available to the readers, rather than in the manuscript text. At the moment, the main take home messages result rather diluted within the details in the text and difficult to identify. The long text (with recurrent names of the genes) results a bit repetitive, difficult to read with the risk of losing interest.
Response:
Thank you for your careful review and valuable suggestions. The content in lines 208-215 of the previous manuscript has been rewritten in lines 234-236 of the revised manuscript, an attached table was added and named Table S3. The content in lines 342-367 of the previous manuscript has been rewritten in lines 368-389 of the revised manuscript, an attached table was added and named Table S7. The content in lines 443-461 of the previous manuscript has been rewritten in lines 463-478 of the revised manuscript, an attached table was added and named Table S11.
3. English: some sentence do not make very much sense to me. Among others: lines 43-46; line 85: “understanding for the function” should be changed into “a better understanding of the function”; line 24: the comprehensive analysis. “the” should be changed into “a comprehensive analysis”.
Response:
Thank you very much for your careful review and helpful advice. The sentence in lines 43-46 in the previous manuscript was used to illustrate that gene expression changes were correlative with polyploidization. In our results, the expression patterns of the GH3 genes from ancestor species changed after allopolyploidization, which just showed that gene expression changes were correlative with polyploidization. “understanding for the function” was replaced by “a better understanding of the function” in lines 87-88 of the revised manuscript. “the comprehensive analysis” was replaced by “a comprehensive analysis” in line 24 of the revised manuscript.
4. Abstract: line 14: “the effect of allopolyplodization on GH3…”I am surprised that the authors only mentioned allopolyploidization while in their conclusions they state that most gene loss occurred after WGT. Was that finding already reported? If not I would rephrase as “the effect of WGT and of allopolyplodization on GH3…”
Response:
Thank you very much for your valuable suggestions. The main purpose of this article is to explore the effects of allopolyploidization on GH3 gene structures and expression levels. Therefore, we identified the GH3 orthologous genes between B. napus and ancestor species, and then focused on describing the gene structures and expression levels between the orthologous pairs. When we studied the changes in the number of GH3 genes, we mentioned the WGT and made a conclusion, which is only a small part of the content.
5. Introduction: Line 61-62. It is stated that group 3 is Arabidopsis specific but this can´t be true if it is later found at least also in the 3 brassicas analysed. Line 70-71: how about apple and legumes as cited in line 535?
Response:
Thank you for your careful review and precious advice.
The sentences in lines 61-62 of the previous manuscript has been rewritten in lines 63-64 of the revised manuscript. The sentence in line 70-71 of the previous manuscript was added “and so on” in line 73 of the revised manuscript.
6. Materials and methods: line 99 “Plants were grown in natural environment” is definitively too vague and should be described more into details especially with regards to the RNA seq analyses performed. In 2.8 there is no trace of how plants were grown, which material was harvested, how, when during development, at which time of the day etc..all variables that highly influence gene expression especially when comparisons want to be done. In 2.7, for the search of cis-elements, standard 2000 bp of upstream genomic seq were considered for all the species. However, research in this paper shows that intragenic and possibly intergenic regions were expanded in B. napus compared with ancestors. Perhaps the whole intergenic region until the upstream genes should be taken in consideration. Moreover, regulatory elements can also be present within introns and in intergenic regions downstream of the genes but this was not taken in consideration.
Response:
Thank you very much for your helpful comments and suggestions. The sentence in line 99 of the previous manuscript has been rewritten in lines 101-104 of the revised manuscript. The details with regards to the RNA seq analyses performed was added in lines 155-171 of the revised manuscript. The condition of the experimental materials was added at lines 155-158 of the revised manuscript. According to the published articles of the GH3 gene family, the studies of the cis-elements of the GH3 genes in the promoter regions were mainly focused on 1500-2000 bp relative to the translation start codon. The maximum value was taken in consideration, and 2000 bp was used in our research. The current studies about cis-elements of GH3 genes are focused on promoter regions, so the cis-elements located in the promoter regions were analyzed in our study, and the promoter region is usually located upstream of the translation start codon, and regulatory elements within introns and intergenic regions downstream of the genes were not taken in consideration.
7. Result 3.1 when talking about the lack of copies (e.g. AtGH3-4 and -13) does this mean that there are also no pseudogenes? I think it would be valuable information to know whether the gene is there in the same syntenic position but it is not functional anymore (especially with regard to discussion line 552-560 and at the end of 3.2). Moreover, in the case of pseudogenes, I would not say that a gene disappear. I think this would be misleading.
Response:
Thank you very much for your critical comments. Maybe there are pseudogenes. The absence of AtGH3-4 and -13 may also met these three possibilities (lines 564-572). We found some genes in the same syntenic positions, but the proteins that these gene corresponded to had incomplete GH3 functional domains, so these genes were discarded, which resulted in the lack of copies. In the case of pseudogenes, the use of “disappear” is indeed inappropriate, the relevant content has been modified in line 556 and 565 of the revised manuscript.
8. 1) Result 3.2 line 205-206:“most of the GH3 genes remain intact”. This chapter it is not about the gene but about the synthenic region, I guess. line 207: not sure where the 13 number came from (my calculation was 38 minus 33 from An-Ar and 25 minus 20 from Cn-Co, so 10?).
2) Lines 208-215: if I understood correctly it is suggested that an incomplete GH3 domain became a complete one after allopolyploidization? I would rather argue that the GH3 domain in B. rapa or B. oleracea has become incomplete in more recent times than when the allopolyploidization took place, whereas it has been kept complete in B. napus.
3) Line 221-224: very unclear to me what the authors want to tell. Is it about this study? About common knowledge? Former literature? Than it should be cited and should be made it clearer. In the case of common knowledge, the use of the past tense “syntenic genes were..” would be wrong then. Present should be used. The same in line 233.
4) Line 228: “demonstrated”, I would say “confirmed”
Response:
Thank you very much for your careful review and helpful advice.
1) “Most of the GH3 genes remain intact” was used to explain that “most of the orthologous GH3 gene pairs (82.6%) between B. rapa and B. oleracea were retained as homologous gene pairs in B. napus”. Genes with orthologous relationships from ancestral species remained intact (no fragments were lost), and then they were able to remain homologous in B. napus and became homologous gene pairs. Table S2 listed the orthologous relationships among the three species, in which four BnaGH3 genes on the A subgenome had no corresponding BraGH3 orthologous gene, and nine BnaGH3 genes on the C subgenome had no corresponding BolGH3 orthologous gene. This sentence has been added in lines 231-233 of the revised manuscript.
2) The content in lines 208-215 of the previous manuscript has been rewritten in lines 234-243 of the revised manuscript.
3) The content in lines 221-224 is common knowledge and has been changed to present tense in lines 249-252 of the revised manuscript. The sentence in line 233 of the previous manuscript was our results, and past tense was used.
4) “Demonstrated” has been replaced by “confirmed” in line 256 of the revised manuscript.
9. Results 3.5 which data has been used for the gene structure analyses? Expression data already available? Was the structure confirmed using the RNA seq performed in this research?
Response:
Thank you very much for your careful review and helpful advice. The GFF files were used for the gene structure analyses. These files contain information on the number and location of exons and introns and can be downloaded from BRAD database. Expression data was added in Table S12, and raw data has been deposited to NCBI database (accession number SRR7816633-SRR7816668). The RNA seq performed was transcriptome sequencing and used to study the gene expression. The structures of all genes have been studied by the previous researchers, and related information is uploaded to the BRAD database. The files containing the gene structure information were GFF files. These GFF files were used to analyze the gene structure in our research.
10. Discussion 4.1 I find strange expression such as line 542 “ the loss of most GH3 genes occurred in the process of Brassica genome triplication”. I would think that the loss happened somewhat after the WGT, not during or in the process. Or again line 548 “accompanied by gene loss” I would think “followed by gene loss”?. I would suggest the author to make a graphic representation of the timing of events WGT and allopolyploidization and corresponding genes that were lost or that arise in in the different time. It would help the reader to understand.
Response:
Thank you very much for your careful review and valuable suggestions. The sentence “the loss of most GH3 genes occurred in the process of Brassica genome triplication” has been changed to “the loss of most GH3 genes happened after Brassica genome triplication” in lines 553-554 of the revised manuscript. “accompanied by gene loss” was replaced by “followed by gene loss” in line 559 of the revised manuscript. The figure with the time of WGT and allopolyploidization events and corresponding genes that were lost or gained in the different time was made and named Figure S3.
11. 1) Discussion 4.1. I wonder how much variation in GH3 there is already at the species level rather than at the interspecific level among the three species analysed here (which could lead to incorrect conclusions). In Arabidopsis several accessions have been sequenced. Is there variation in GH3 genes within the same species? Perhaps a disclaimer should be made on this point.
2) Line 534-535: it should make clearer that the report from the previous researcher was from other species.
Response:
Thank you very much for your careful review.
1) So far, for Arabidopsis thaliana, the GH3 gene family has not been compared among its accessions. It is speculated that the GH3 domain should be conserved, and the gene structures may also be relatively conservative, but expression levels may vary between its accessions.
2) It has been made clear that the report from the previous researcher was from legumes and apple in lines 544-546 of the revised manuscript.
Round 2
Reviewer 1 Report
With the revisions, the manuscript is much improved. Especially updating the heatmaps and including Z-score calculations help to understand the analysis.